# Neurally-Guided Procedural Models: Amortized Inference for Procedural Graphics Programs using Neural Networks

**Daniel Ritchie**
Stanford University

**Anna Thomas**
Stanford University

**Pat Hanrahan**
Stanford University

**Noah D. Goodman**
Stanford University

## Abstract

Probabilistic inference algorithms such as Sequential Monte Carlo (SMC) provide powerful tools for constraining procedural models in computer graphics, but they require many samples to produce desirable results. In this paper, we show how to create procedural models which *learn* how to satisfy constraints. We augment procedural models with neural networks which control how the model makes random choices based on the output it has generated thus far. We call such models *neurally-guided procedural models*. As a pre-computation, we train these models to maximize the likelihood of example outputs generated via SMC. They are then used as efficient SMC importance samplers, generating high-quality results with very few samples. We evaluate our method on L-system-like models with image-based constraints. Given a desired quality threshold, neurally-guided models can generate satisfactory results up to 10x faster than unguided models.

## 1  Introduction

Procedural modeling, or the use of randomized procedures to generate computer graphics, is a powerful technique for creating visual content. It facilitates efficient content creation at massive scale, such as procedural cities [13]. It can generate fine detail that would require painstaking effort to create by hand, such as decorative floral patterns [24]. It can even generate surprising or unexpected results, helping users to explore large or unintuitive design spaces [19].

Many applications demand control over procedural models: making their outputs resemble examples [22, 2], fit a target shape [17, 21, 20], or respect functional constraints such as physical stability [19]. Bayesian inference provides a general-purpose control framework: the procedural model specifies a generative prior, and the constraints are encoded as a likelihood function. Posterior samples can then be drawn via Markov Chain Monte Carlo (MCMC) or Sequential Monte Carlo (SMC). Unfortunately, these algorithms often require many samples to converge to high-quality results, limiting their usability for interactive applications. Sampling is challenging because the constraint likelihood implicitly defines complex (often non-local) dependencies not present in the prior. Can we instead make these dependencies *explicit* by encoding them in a model's generative logic? Such an explicit model could simply be run forward to generate high-scoring results.

In this paper, we propose an amortized inference method for learning an approximation to this perfect explicit model. Taking inspiration from recent work in amortized variational inference, we augment the procedural model with neural networks that control how the model makes random choices based on the partial output it has generated. We call such a model a *neurally-guided procedural model*. We train these models by maximizing the likelihood of example outputs generated via SMC using a large number of samples, as an offline pre-process. Once trained, they can be used as efficient SMC importance samplers. By investing time up-front generating and training on many examples, our system effectively 'pre-compiles' an efficient sampler that can generate further results much faster.

For a given likelihood threshold, neurally-guided models can generate results which reliably achieve that threshold using 10-20x fewer particles and up to 10x less compute time than an unguided model.

In this paper, we focus on *accumulative* procedural models that repeatedly add new geometry to a structure. For our purposes, a procedural model is accumulative if, while executing, it provides a 'current position' $\mathbf{p}$ from which geometry generation will continue. Many popular growth models, such as L-systems, are accumulative [16]. We focus on 2D models ($\mathbf{p} \in \mathbb{R}^2$) which generate images, though the techniques we present extend naturally to 3D.

## 2 Related Work

**Guided Procedural Modeling**    Procedural models can be guided using non-probabilistic methods. Open L-systems can query their spatial position and orientation, allowing them to prune their growth to an implicit surface [17, 14]. Recent follow-up work supports larger models by decomposing them into separate regions with limited interaction [1]. These methods were specifically designed to fit procedural models to shapes. In contrast, our method *learns* how to guide procedural models and is generally applicable to constraints expressable as likelihood functions.

**Generatively Capturing Dependencies in Procedural Models**    A recent system by Dang et al. modifies a procedural grammar so that its output distribution reflects user preference scores given to example outputs [2]. Like us, they use generative logic to capture dependencies induced by a likelihood function (in their case, a Gaussian process regression over user-provided examples). Their method splits non-terminal symbols in the original grammar, giving it more combinatorial degrees of freedom. This works well for discrete dependencies, whereas our method is better suited for continuous constraint functions, such as shape-fitting.

**Neural Amortized Inference**    Our method is also inspired by recent work in amortized variational inference using neural variational families [11, 18, 8], but it uses a different learning objective. Prior work has also aimed to train efficient neural SMC importance samplers [5, 15]. These efforts focused on time series models and Bayesian networks, respectively; we focus on a class of structured procedural models, the characteristics of which permit different design decisions:

- The likelihood of a partially-generated output can be evaluated at any time and is a good heuristic for the likelihood of the completed output. This is different from e.g. time series models, where the likelihood at each step considers a previously-unseen data point.
- They make many local random choices but have no global/top-level parameters.
- They generate images, which naturally support coarse-to-fine feature extraction.

These properties informed the design of our neurally-guided model architecture.

## 3 Approach

Consider a simple procedural modeling program `chain` that recursively generates a random sequence of linear segments, constrained to match a target image. Figure 1a shows the text of this program, along with samples generated from it (drawn in black) against several target images (drawn in gray). Chains generated by running the program forward do not match the targets, since forward sampling is oblivious to the constraint. Instead, we can generate constrained samples using Sequential Monte Carlo (SMC) [20]. This results in final chains that more closely match the target images. However, the algorithm requires many particles—and therefore significant computation—to produce acceptable results. Figure 1a shows that $N = 10$ particles is not sufficient.

In an ideal world, we would not need costly inference algorithms to generate constraint-satisfying results. Instead, we would have access to an 'oracle' program, `chain_perfect`, that perfectly fills in the target image when run forward. While such an oracle can be difficult or impossible to write by hand, it is possible to *learn* a program `chain_neural` that comes close. Figure 1b shows our approach. For each random choice in the program text (e.g. `gaussian`, `flip`), we replace the parameters of that choice with the output of a neural network. This neural network's inputs (abstracted as "...") include the target image as well the partial output image the program has generated thus far. The network thus

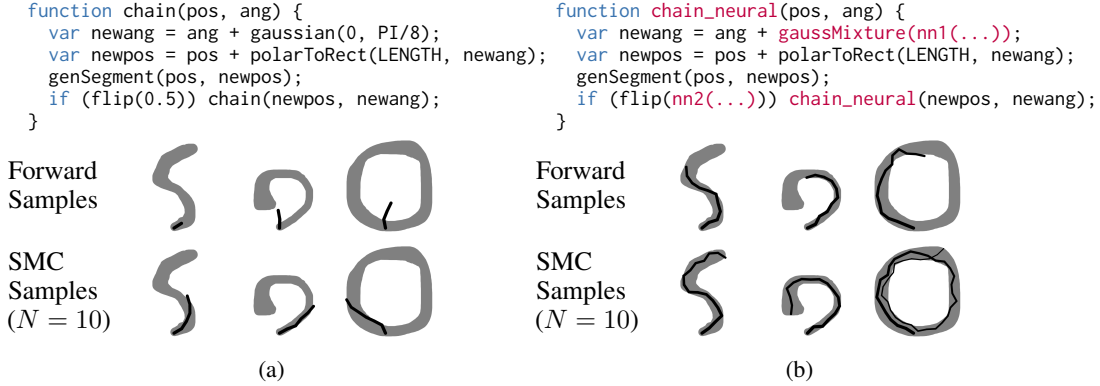

```
function chain(pos, ang) {                              function chain_neural(pos, ang) {
    var newang = ang + gaussian(0, PI/8);                   var newang = ang + gaussMixture(nn1(...));
    var newpos = pos + polarToRect(LENGTH, newang);         var newpos = pos + polarToRect(LENGTH, newang);
    genSegment(pos, newpos);                                genSegment(pos, newpos);
    if (flip(0.5)) chain(newpos, newang);                   if (flip(nn2(...))) chain_neural(newpos, newang);
}                                                       }
```

Forward Samples

SMC Samples ($N = 10$)

(a)

Forward Samples

SMC Samples ($N = 10$)

(b)

Figure 1: Turning a linear chain model into a neurally-guided model. *(a)* The original program. When outputs (shown in black) are constrained to match a target image (shown in gray), SMC requires many particles to achieve good results. *(b)* The neurally-guided model, where random choice parameters are computed via neural networks. Once trained, forward sampling from this model adheres closely to the target image, and SMC with only 10 particles consistently produces good results.

shapes the distribution over possible choices, guiding the programs's future output based on the target image and its past output. These neural nets affect both continuous choices (e.g. angles) as well as control flow decisions (e.g. recursion): they dictate where the chain goes next, as well as whether it keeps going at all. For continuous choices such as gaussian, we also modify the program to sample from a mixture distribution. This helps the program handle situations where the constraints permit multiple distinct choices (e.g. in which direction to start the chain for the circle-shaped target image in Figure 1).

Once trained, chain_neural generates constraint-satisfying results more efficiently than its un-guided counterpart. Figure 1b shows example outputs: forward samples adhere closely to the target images, and SMC with 10 particles is sufficient to produce chains that fully fill the target shape. The next section describes the process of building and training such neurally-guided procedural models.

## 4   Method

For our purposes, a *procedural model* is a generative probabilistic model of the following form:

$$P_{\mathbf{M}}(\mathbf{x}) = \prod_{i=1}^{|\mathbf{x}|} p_i(\mathbf{x}_i; \Phi_i(\mathbf{x}_1, \ldots, \mathbf{x}_{i-1}))$$

Here, $\mathbf{x}$ is the vector of random choices the procedural modeling program makes as it executes. The $p_i$'s are local probability distributions from which each successive random choice is drawn. Each $p_i$ is parameterized by a set of parameters (e.g. mean and variance, for a Gaussian distribution), which are determined by some function $\Phi_i$ of the previous random choices $\mathbf{x}_1, \ldots, \mathbf{x}_{i-1}$.

A *constrained procedural model* also includes an unnormalized likelihood function $\ell(\mathbf{x}, \mathbf{c})$ that measures how well an output of the model satisfies some constraint $\mathbf{c}$:

$$P_{\mathbf{CM}}(\mathbf{x}|\mathbf{c}) = \frac{1}{Z} \cdot P_{\mathbf{M}}(\mathbf{x}) \cdot \ell(\mathbf{x}, \mathbf{c})$$

In the chain example, $\mathbf{c}$ is the target image, with $\ell(\cdot, \mathbf{c})$ measuring similarity to that image.

A *neurally-guided procedural model* modifies a procedural model by replacing each parameter function $\Phi_i$ with a neural network:

$$P_{\mathbf{GM}}(\mathbf{x}|\mathbf{c}; \theta) = \prod_{i=1}^{|\mathbf{x}|} \tilde{p}_i(\mathbf{x}_i; \mathrm{NN}_i(I(\mathbf{x}_1, \ldots, \mathbf{x}_{i-1}), \mathbf{c}; \theta))$$

where $I(\mathbf{x}_1, \ldots, \mathbf{x}_{i-1})$ renders the model output after the first $i - 1$ random choices, and $\theta$ are the network parameters. $\tilde{p}_i$ is a mixture distribution if random choice $i$ is continuous; otherwise, $\tilde{p}_i = p_i$.

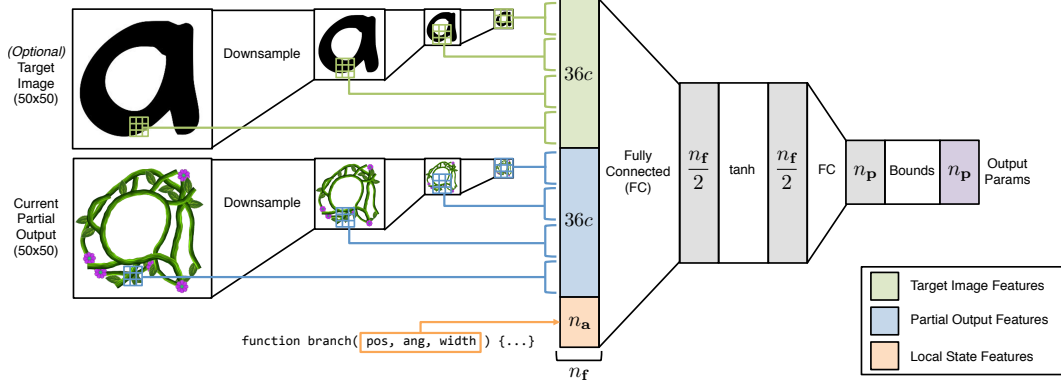

Figure 2: Network architecture for neurally-guided procedural models. The outputs are the parameters for a random choice probability distribution. The inputs come from three sources: *Local State Features* are the arguments to the function in which the random choice occurs; *Partial Output Features* come from 3x3 pixel windows of the partial image the model has generated, extracted at multiple resolutions, around the procedural model's current position; *Target Image Features* are analogous windows extracted from the target image, if the constraint requires one.

To train a neurally-guided procedural model, we seek parameters $\theta$ such that $P_{\mathbf{GM}}$ is as close as possible to $P_{\mathbf{CM}}$. This goal can be formalized as minimizing the conditional KL divergence $D_{\mathrm{KL}}(P_{\mathbf{CM}}||P_{\mathbf{GM}})$ (see the supplemental materials for derivation):

$$\min_{\theta} D_{\mathrm{KL}}(P_{\mathbf{CM}}||P_{\mathbf{GM}}) \approx \max_{\theta} \frac{1}{N} \sum_{s=1}^{N} \log P_{\mathbf{GM}}(\mathbf{x}_s|\mathbf{c}_s;\theta) \qquad \mathbf{x}_s \sim P_{\mathbf{CM}}(\mathbf{x}), \; \mathbf{c}_s \sim P(\mathbf{c}) \quad (1)$$

where the $\mathbf{x}_s$ are example outputs generated using SMC, given a $\mathbf{c}_s$ drawn from some distribution $P(\mathbf{c})$ over constraints, e.g. uniform over a set of training images. This is simply maximizing the likelihood of the $\mathbf{x}_s$ under the neurally-guided model. Training then proceeds via stochastic gradient ascent using the gradient

$$\nabla \log P_{\mathbf{GM}}(\mathbf{x}|\mathbf{c};\theta) = \sum_{i=1}^{|\mathbf{x}|} \nabla \log \tilde{p}_i(\mathbf{x}_i; \mathrm{NN}_i(I(\mathbf{x}_1, \ldots, \mathbf{x}_{i-1}), \mathbf{c}; \theta)) \qquad (2)$$

The trained $P_{\mathbf{GM}}(\mathbf{x}|\mathbf{c};\theta)$ can then be used as an importance distribution for SMC.

It is worth noting that using the other direction of KL divergence, $D_{\mathrm{KL}}(P_{\mathbf{GM}}||P_{\mathbf{CM}})$, leads to the marginal likelihood lower bound objective used in many black-box variational inference algorithms [23, 6, 11]. This objective requires training samples from $P_{\mathbf{GM}}$, which are much less expensive to generate than samples from $P_{\mathbf{CM}}$. When used for procedural modeling, however, it leads to models whose outputs lack diversity, making them unsuitable for generating visually-varied content. This behavior is due to a well-known property of the objective: minimizing it produces approximating distributions that are overly-compact, i.e. concentrating their probability mass in a smaller volume of the state space than the true distribution being approximated [10]. Our objective is better suited for training proposal distributions for importance sampling methods (such as SMC), where the target density must be absolutely continuous with respect to the proposal density [3].

## 4.1 Neural Network Architecture

Each network $\mathrm{NN}_i$ should predict a distribution over choice $i$ that is as close as possible to its true posterior distribution. More complex networks capture more dependencies and increase accuracy but require more computation time to execute. We can also increase accuracy at the cost of computation time by running SMC with more particles. If our networks are too complex (i.e. accuracy provided per unit computation time is too low), then the neurally-guided model $P_{\mathbf{GM}}$ will be outperformed by simply using more particles with the original model $P_{\mathbf{M}}$. For neural guidance to provide speedups, we require networks that pack as much predictive power into as simple an architecture as possible.

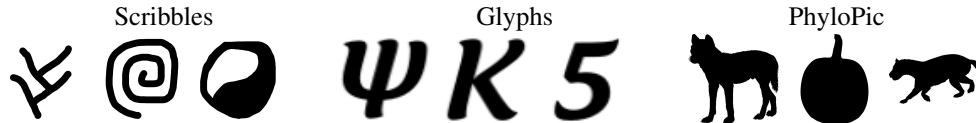

Figure 3: Example images from our datasets.

Figure 2 shows our network architecture: a multilayer perceptron with $n_\mathbf{f}$ inputs, one hidden layer of size $n_\mathbf{f}/2$ with a $\mathtt{tanh}$ nonlinearity, and $n_\mathbf{p}$ outputs, where $n_\mathbf{p}$ is the number of parameters the random choice expects. We found that a simpler linear model did not perform as well per unit time. Since some parameters are bounded (e.g. Gaussian variance must be positive), each output is remapped via an appropriate bounding transform (e.g. $e^x$ for non-negative parameters). The inputs come from several sources, each providing the network with decision-critical information:

**Local State Features**  The model's current position $\mathbf{p}$, the current orientation of any local reference frame, etc. We access this data via the arguments of the function call in which the random choice occurs, extracting all $n_\mathbf{a}$ scalar arguments and normalizing them to $[-1, 1]$.

**Partial Output Features**  Next, the network needs information about the output the model has already generated. The raw pixels of the current partial output image $I(\cdot)$ provide too much data; we need to summarize the relevant image contents. We extract 3x3 windows of pixels around the model's current position $\mathbf{p}$ at four different resolution levels, with each level computed by downsampling the previous level via a 2x2 box filter. This results in $36c$ features for a $c$-channel image. This architecture is similar to the foveated 'glimpses' used in visual attention models [12]. Convolutional networks might also be used here, but this approach provided better performance per unit of computation time.

**Target Image Features**  Finally, if the constraint being enforced involves a target image, as in the chain example of Section 3, we also extract multi-resolution windows from this image. These additional $36c$ features allow the network to make appropriate decisions for matching the image.

### 4.2  Training

We train with stochastic gradient ascent (see Equation 2). We use the Adam algorithm [7] with $\alpha = \beta = 0.75$, step size 0.01, and minibatch size one. We terminate training after 20000 iterations.

## 5  Experiments

In this section, we evaluate how well neurally-guided procedural models capture image-based constraints. We implemented our prototype system in the WebPPL probabilistic programming language [4] using the adnn neural network library.[1] All timing data was collected on an Intel Core i7-3840QM machine with 16GB RAM running OSX 10.10.5.

### 5.1  Image Datasets

In experiments which require target images, we use the following image collections:

- **Scribbles**: 49 binary mask images drawn by hand with the brush tool in Photoshop. Includes shapes with thick and thin regions, high and low curvature, and self-intersections.
- **Glyphs**: 197 glyphs from the FF Tartine Script Bold typeface (all glyphs with only one foreground component and at least 500 foreground pixels when rendered at 129x97).
- **PhyloPic**: 35 images from the PhyloPic silhouette database.[2]

We augment the dataset with a horizontally-mirrored copy of each image, and we annotate each image with a starting point and direction from which to initialize the execution of a procedural model. Figure 3 shows some representative images from each collection.

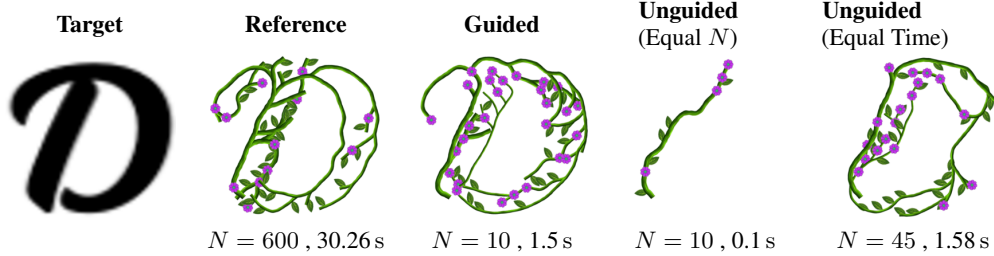

| Target | Reference | Guided | Unguided (Equal $N$) | Unguided (Equal Time) |
|--------|-----------|--------|----------------------|------------------------|
| | $N = 600$ , $30.26\,\text{s}$ | $N = 10$ , $1.5\,\text{s}$ | $N = 10$ , $0.1\,\text{s}$ | $N = 45$ , $1.58\,\text{s}$ |

Figure 4: Constraining a vine-growth procedural model to match a target image. $N$ is the number of SMC particles used. *Reference* shows an example result after running SMC on the unguided model with a large number of particles. Neurally-guided models generate results of this quality in a couple seconds; the unguided model struggles given the same amount of particles or computation time.

## 5.2 Shape Matching

We first train neurally-guided procedural models to match 2D shapes specified as binary mask images. If $\mathcal{D}$ is the spatial domain of the image, then the likelihood function for this constraint is

$$\ell_{\textbf{shape}}(\mathbf{x}, \mathbf{c}) = \mathcal{N}\left(\frac{\text{sim}(I(\mathbf{x}), \mathbf{c}) - \text{sim}(\mathbf{0}, \mathbf{c})}{1 - \text{sim}(\mathbf{0}, \mathbf{c})}, 1, \sigma_{\textbf{shape}}\right) \tag{3}$$

$$\text{sim}(I_1, I_2) = \frac{\sum_{\mathbf{p} \in \mathcal{D}} w(\mathbf{p}) \cdot \mathbb{1}\{I_1(\mathbf{p}) = I_2(\mathbf{p})\}}{\sum_{\mathbf{p} \in \mathcal{D}} w(\mathbf{p})} \qquad w(\mathbf{p}) = \begin{cases} 1 & \text{if } I_2(\mathbf{p}) = 0 \\ 1 & \text{if } ||\nabla I_2(\mathbf{p})|| = 1 \\ w_{\textbf{filled}} & \text{if } ||\nabla I_2(\mathbf{p})|| = 0 \end{cases}$$

where $\nabla I(\mathbf{p})$ is a binary edge mask computed using the Sobel operator. This function encourages the output image $I(\mathbf{x})$ to be similar to the target mask $\mathbf{c}$, where similarity is normalized against $\mathbf{c}$'s similarity to an empty image $\mathbf{0}$. Each pixel $\mathbf{p}$'s contribution is weighted by $w(\mathbf{p})$, determined by whether the target mask is empty, filled, or has an edge at that pixel. We use $w_{\textbf{filled}} = 0.\bar{6}$, so empty and edge pixels are worth 1.5 times more than filled pixels. This encourages matching of perceptually-important contours and negative space. $\sigma_{\textbf{shape}} = 0.02$ in all experiments.

We wrote a WebPPL program which recursively generates vines with leaves and flowers and then trained a neurally-guided version of this program to capture the above likelihood. The model was trained on 10000 example outputs, each generated using SMC with 600 particles. Target images were drawn uniformly at random from the Scribbles dataset. Each example took on average 17 seconds to generate; parallelized across four CPU cores, the entire set of examples took approximately 12 hours to generate. Training took 55 minutes in our single-threaded implementation.

Figure 4 shows some outputs from this program. 10-particle SMC produces recognizable results with the neurally-guided model (*Guided*) but not with the unguided model (*Unguided (Equal $N$)*). A more equitable comparison is to give the unguided model the same amount of wall-clock time as the guided model. While the resulting outputs fare better, the target shape is still obscured (*Unguided (Equal Time)*). We find that the unguided model needs ~200 particles to reliably match the performance of the guided model. Additional results are shown in the supplemental materials.

Figure 5 shows a quantitative comparison between five different models on the shape matching task:

- **Unguided**: The original, unguided procedural model.
- **Constant Params**: The neural network for each random choice is a vector of constant parameters (i.e. a *partial mean field approximation* [23]).
- **+ Local State Features**: Adding the local state features described in Section 4.1.
- **+ Target Image Features**: Adding the target image features described in Section 4.1.
- **All Features**: Adding the partial output features described in Section 4.1.

We test each model on the Glyph dataset and report the median normalized similarity-to-target achieved (i.e. argument one to the Gaussian in Equation 3), plotted in Figure 5a. The performance

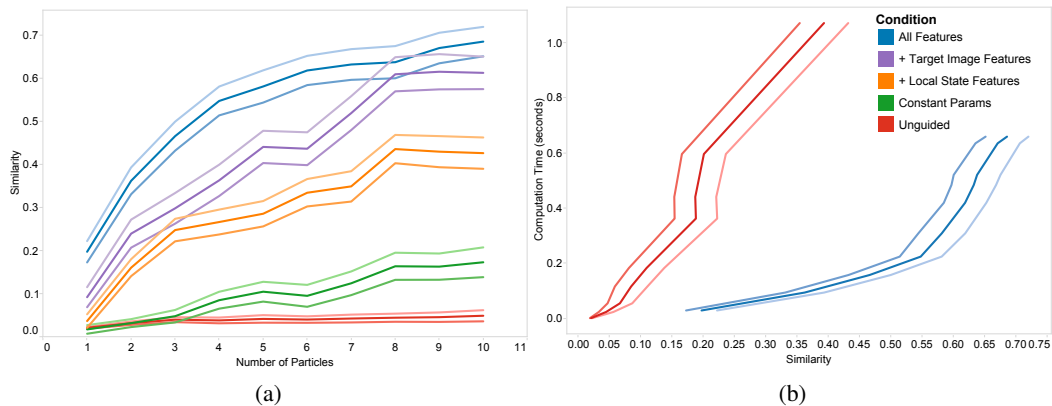

(a)                                                           (b)

Figure 5: Shape matching performance comparison. *"Similarity"* is median normalized similarity to target, averaged over all targets in the test dataset. Bracketing lines show 95% confidence bounds. *(a)* Performance as number of SMC particles increases. The neurally-guided model achieves higher average similarity as more features are added. *(b)* Computation time required as desired similarity increases. The vertical gap between the two curves indicates speedup (which can be up to 10x).

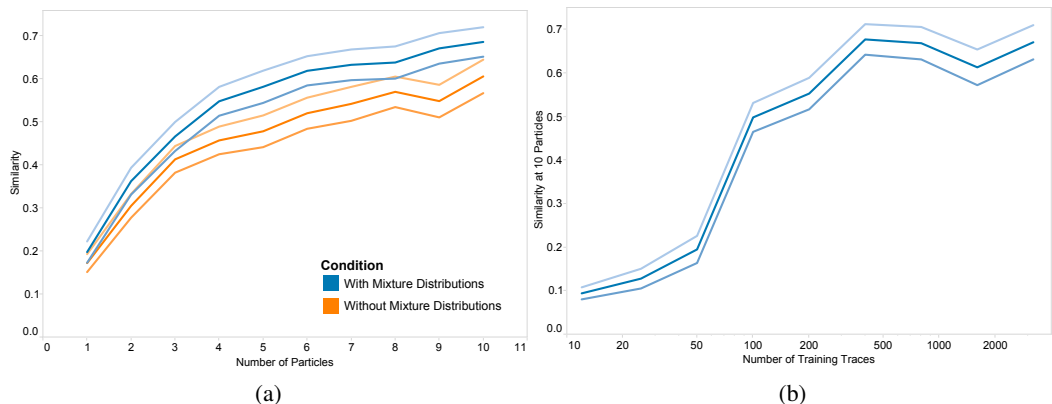

(a)                                                           (b)

Figure 6: *(a)* Using four-component mixtures for continuous random choices boosts performance. *(b)* The effect of training set size on performance (at 10 SMC particles), plotted on a logarithmic scale. Average similarity-to-target levels off at ∼1000 examples.

of the guided model improves with the addition of more features; at 10 particles, the full model is already approaching an asymptote. Figure 5b shows the wall-clock time required to achieve increasing similarity thresholds. The vertical gap between the two curves shows the speedup given by neural guidance, which can be as high as 10x. For example, the + *Local State Features* model reaches similarity 0.35 about 5.5 times faster than the *Unguided* model, the + *Target Image Features* model is about 1.5 times faster still, and the *All Features* Model is about 1.25 times faster than that. Note that we trained on the Scribbles dataset but tested on the Glyphs dataset; these results suggest that our models can generalize to qualitatively-different previously-unseen images.

Figure 6a shows the benefit of using mixture guides for continuous random choices. The experimental setup is the same as in Figure 5. We compare a model which uses four-component mixtures with a no-mixture model. Using mixtures boosts performance, which we alluded to in Section 3: at shape intersections, such as the crossing of the letter 't,' the model benefits from multimodal uncertainty. Using more than four mixture components did not improve performance on this test dataset.

We also investigate how the number of training examples affects performance. Figure 6b plots the median similarity at 10 particles as training set size increases. Performance increases rapidly for the first few hundred examples before leveling off, suggesting that ∼1000 sample traces is sufficient (for our particular choice of training set, at least). This may seem surprising, as many published neurally-based learning systems require many thousands to millions of training examples. In our case,

| Reference $N = 600$ | Guided $N = 15$ | Unguided (Equal $N$) | Unguided (Equal Time) |
|---|---|---|---|

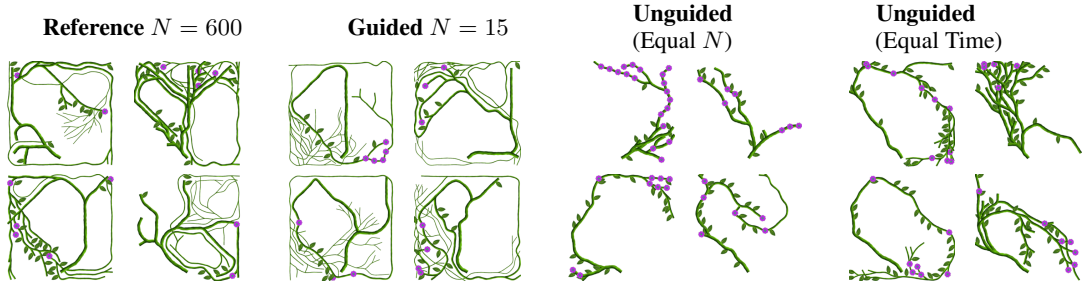

Figure 7: Constraining the vine-growth program to generate circuit-like patterns. *Reference* outputs took around $\sim 70$ seconds to generate; outputs from the guided model took $\sim 3.5$ seconds.

each training example contains hundreds to thousands of random choices, each of which provides a learning signal—in this way, the training data is "bigger" than it appears. Our implementation generates 1000 samples in just over an hour using four CPU cores.

### 5.3 Stylized "Circuit" Design

We next train neurally-guided procedural models to capture a likelihood that does not use a target image: constraining the vines program to resemble a stylized circuit design. To achieve the dense packing of long wire traces that is one of the most striking visual characteristics of circuit boards, we encourage a percentage $\tau$ of the image to be filled ($\tau = 0.5$ in our results) and to have a dense, high-magnitude gradient field, as this tends to create many long rectilinear or diagonal edges:

$$\ell_{\mathbf{circ}}(\mathbf{x}) = \mathcal{N}\big(\text{edge}(I(\mathbf{x})) \cdot (1 - \eta(\text{fill}(I(\mathbf{x})), \tau)), 1, \sigma_{\mathbf{circ}}\big) \tag{4}$$

$$\text{edge}(I) = \frac{1}{|\mathcal{D}|} \sum_{\mathbf{p} \in \mathcal{D}} ||\nabla I(\mathbf{p})|| \quad \text{fill}(I) = \frac{1}{|\mathcal{D}|} \sum_{\mathbf{p} \in \mathcal{D}} I(\mathbf{p})$$

where $\eta(x, \bar{x})$ is the relative error of $x$ from $\bar{x}$ and $\sigma_{\mathbf{circ}} = 0.01$. We also penalize geometry outside the bounds of the image, encouraging the program to fill in a rectangular "die"-like region. We train on 2000 examples generated using SMC with 600 particles. Example generation took 10 hours and training took under two hours. Figure 7 shows outputs from this program. As with shape matching, the neurally-guided model generates high-scoring results significantly faster than the unguided model.

## 6 Conclusion and Future Work

This paper introduced neurally-guided procedural models: constrained procedural models that use neural networks to capture constraint-induced dependencies. We showed how to train guides for accumulative models with image-based constraints using a simple-yet-powerful network architecture. Experiments demonstrated that neurally-guided models can generate high-quality results significantly faster than unguided models.

Accumulative procedural models provide a current position $\mathbf{p}$, which is not true of other generative paradigms (e.g. texture generation, which generates content across its entire spatial domain). In such settings, the guide might instead *learn* what parts of the current partial output are relevant to each random choice using an attention process [12].

Using neural networks to predict random choice parameters is just one possible program transformation for generatively capturing constraints. Other transformations, such as control flow changes, may be necessary to capture more types of constraints. A first step in this direction would be to combine our approach with the grammar-splitting technique of Dang et al. [2].

Methods like ours could also accelerate inference for other applications of procedural models, e.g. as priors in analysis-by-synthesis vision systems [9]. A robot perceiving a room through an onboard camera, detecting chairs, then fitting a procedural model to the detected chairs could learn importance distributions for each step of the chair-generating process (e.g. the number of parts, their size, arrangement, etc.) Future work is needed to determine appropriate neural guides for such domains.

## Footnotes

[1]https://github.com/dritchie/adnn

[2]http://phylopic.org

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
