[Supplementary Material]

# Supplemental Materials for Paper 340 Neurally-Guided Procedural Models: Amortized Inference for Procedural Graphics Programs using Neural Networks

## 1 Derivation of Equation 1

$$\min_{\theta} D_{\mathrm{KL}}(P_{\mathbf{CM}}||P_{\mathbf{GM}})$$

$$= \min_{\theta} \mathbb{E}_{P(\mathbf{c})}\left[\mathbb{E}_{P_{\mathbf{CM}}(\mathbf{x}|\mathbf{c})}\left[\log \frac{P_{\mathbf{CM}}(\mathbf{x}|\mathbf{c})}{P_{\mathbf{GM}}(\mathbf{x}|\mathbf{c};\theta)}\right]\right]$$

$$= \min_{\theta} \mathbb{E}_{P(\mathbf{c})}\left[\mathbb{E}_{P_{\mathbf{CM}}(\mathbf{x}|\mathbf{c})}\left[\log P_{\mathbf{CM}}(\mathbf{x}|\mathbf{c}) - \log P_{\mathbf{GM}}(\mathbf{x}|\mathbf{c};\theta)\right]\right]$$

$$= \max_{\theta} \mathbb{E}_{P(\mathbf{c})}\left[\mathbb{E}_{P_{\mathbf{CM}}(\mathbf{x}|\mathbf{c})}\left[\log P_{\mathbf{GM}}(\mathbf{x}|\mathbf{c};\theta) - \log P_{\mathbf{CM}}(\mathbf{x}|\mathbf{c})\right]\right]$$

$$= \max_{\theta} \mathbb{E}_{P(\mathbf{c})}\left[\mathbb{E}_{P_{\mathbf{CM}}(\mathbf{x}|\mathbf{c})}\left[\log P_{\mathbf{GM}}(\mathbf{x}|\mathbf{c};\theta)\right]\right]$$

$$\approx \max_{\theta} \frac{1}{N}\sum_{s=1}^{N}\log P_{\mathbf{GM}}(\mathbf{x}_s|\mathbf{c}_s;\theta) \qquad \mathbf{x}_s \sim P_{\mathbf{CM}}(\mathbf{x}|\mathbf{c}), \ \mathbf{c}_s \sim P(\mathbf{c})$$

In the second-to-last step, the $\log P_{\mathbf{CM}}(\mathbf{x}|\mathbf{c})$ term is dropped because it does not depend on $\theta$. In the last step, we approximate the expectations with an average over a finite set of samples.

# 2 Additional Results

| 0.99 s | 0.81 s | 1.01 s | 1.03 s | 0.9 s | 1.16 s | 0.86 s | 1.08 s |

Figure 1: Targeting letter shapes with a neurally-guided procedural lightning program. Generated using SMC with 10 particles; compute time required is shown below each letter. Best viewed on a high-resolution display.

Figure 2: Performance comparison for the circuit design problem (section 4.3 in the main paper). *"Score"* is median normalized score (i.e. argument one to the Gaussian in Equation 4 of the main paper), averaged over 50 runs. The neurally-guided version achieves significantly higher average scores than the unguided version given the same number of particles or the same amount of compute time.

| **Target** | **Reference** | **Guided** | **Unguided** (Equal $N$) | **Unguided** (Equal Time) |
|---|---|---|---|---|
| | $N = 600$ , $38.68\,\mathrm{s}$ | $N = 5$ , $0.86\,\mathrm{s}$ | $N = 5$ , $0.09\,\mathrm{s}$ | $N = 30$ , $0.83\,\mathrm{s}$ |
| | $N = 600$ , $33.5\,\mathrm{s}$ | $N = 10$ , $1.23\,\mathrm{s}$ | $N = 10$ , $0.14\,\mathrm{s}$ | $N = 40$ , $1.28\,\mathrm{s}$ |
| | $N = 600$ , $25.55\,\mathrm{s}$ | $N = 15$ , $1.75\,\mathrm{s}$ | $N = 15$ , $0.23\,\mathrm{s}$ | $N = 50$ , $1.73\,\mathrm{s}$ |
| | $N = 600$ , $20.76\,\mathrm{s}$ | $N = 10$ , $0.81\,\mathrm{s}$ | $N = 10$ , $0.15\,\mathrm{s}$ | $N = 40$ , $0.85\,\mathrm{s}$ |
| | $N = 600$ , $25.5\,\mathrm{s}$ | $N = 10$ , $1.04\,\mathrm{s}$ | $N = 10$ , $0.14\,\mathrm{s}$ | $N = 40$ , $1.05\,\mathrm{s}$ |

Figure 3: Additional shape matching results (section 4.2 in the main paper).