[Reviews · NeurIPS 2016]

Reviewer 1

Summary

The authors propose neurally guided procedural models. A procedural model is a stochastic L system implemented as a particular form of constrained probabilistic program, with a (possibly complex) likelihood term. Naive inference in this model struggles. The authors propose a new discriminative inference algorithm, where a trained neural network provides proposals for a sequential monte-carlo algorithm. Results indicate that the neural network significantly improve performance.

Qualitative Assessment

I'm not sure where this paper should land. On the positive side, I felt that this paper was creative, and contributes some interesting technical insights - the combination of DNNs and probabilistic inference is an important topic, and the use KL(pq) plus SMC certainly seems novel and much more general than what this paper would suggest. I think it's also important that the DNN "looks at" both previous trace parameters and the data; this sort of data-driven inference is an important topic in general. On the negative side, I wish this framework were more general. It doesn't seem that L-systems form part of the solution to most ML problems (and the applications seem a bit off-target for NIPS--is this maybe more of a graphics paper?). And while the framework of amortized inference seems reasonable to me, it's unclear how generally applicable it is. I would love to see a more general approach based on blending probabilistic programming and DNNs, although it seems that that will be hard to do; it would be hard to deal with programs with varying number of parameters (especially nonparametrics and other structure modifying distributions). I would put this paper on the bottom of my "accept" pile. I'm inclined to favor it because the themes it deals with are important, but weak applications and lack of generality are damping my enthusiasm.

Confidence in this Review

3-Expert (read the paper in detail, know the area, quite certain of my opinion)


Reviewer 2

Summary

The paper considers neural network parameterized accumulative procedural models and demonstrates that such models can be learned from posterior samples generated by a sequential Monte Carlo algorithm. The learned model is then used generatively at a fraction of the cost of SMC. Experimental results look promising.

Qualitative Assessment

Although the presented approach of generating samples from a simulator (in this case a SMC) and fitting a neural network parametrized model to the generated samples isn't terribly novel, the application to procedural models is interesting, the experiments are thorough for the most part and the results seem promising. I only have a few concerns with the paper: 1) It is interesting that a single hidden layer feed forward network seems to be sufficient for learning the "neurally guided procedural model". An alternate procedural model parametrized by simpler linear models should be much faster to train and it would be interesting to see how it performs. Taking this one step further, it would also be interesting to see a version where the procedural modeling program is parametrized directly by the parameters of interest (instead of the conditional distribution of the parameters being parameterized by a NN). 2) In line 176-177 the authors state --- "The neural network for each random choice is a vector of constant parameters". It is unclear what the authors mean by this and needs further elaboration.

Confidence in this Review

3-Expert (read the paper in detail, know the area, quite certain of my opinion)


Reviewer 3

Summary

This paper is concerned with randomised algorithms that generate computer graphics. It proposes to use a neural network which determines how to make random chouces based on partial outputs.

Qualitative Assessment

This is a well-written paper. I am not an expert in the area but the work seems to be sufficiently novel based on how the paper is situated in the related works. The neural network part is a standard multi-layer perceptron; the design is appropriate for the task. The experimental setup us sound, the results are well presented and the neurally-guided model is compared with a non-guided model. The claims are supported by the results. 1. Why using a neural network and not another learning algorithm? Neural networks (multi-layer perceptrons) are slow to train and sensitive to many parameters. 2. Can you give more details about the practical applications of the proposed algorithm and the significance of the faster training using the neural network?

Confidence in this Review

1-Less confident (might not have understood significant parts)


Reviewer 4

Summary

The authors consider the task of generating samples from constrained procedural models for computer graphics, similar to that considered in the stochastically ordered SMC paper by Ritchie et. al. (as cited by authors). They present an inference procedure based on sequential Monte Carlo, similar to recent approaches proposed by Gu et al. (as cited) and Paige & Wood (see below for ref), in which data-driven importance proposals are learned by gradient descent on an importance sampling estimate of the evidence upper bound (i.e. an expectation-propagation). The main contribution is the proposed neural net architecture for the data-driven proposals, which is specific to procedural graphics problems. This architecture that shows good performance in the experiments under consideration.

Qualitative Assessment

I think this paper is clearly written and makes some reasonable contributions, but could do with an editing pass to frame the approach a bit better and relate it to recent contributions that similarly seek to amortize inference in sequential models by training neural net proposals. While the authors are the first (to my knowledge) to train NN proposals for SMC in a procedural graphics / probabilistic programming setting, their approach is of course very closely related the one developed by Gu and colleagues [1], which the authors cite and the one proposed by Paige and Wood [2], which they do not. I am assuming this work was done more or less concurrently and independently, and I in principle don't see a problem for this paper from the point of view of novelty. That said, the paper in its current revision still reads a bit like a graphics paper that pitches using NN proposals for SMC as its core idea. This is unfortunate, in that it would have been nice to see the authors relate their work to that done by others. When it comes to amortizing inference in sequential problems by training neural nets, the key question (in my opinion) is how the design of the neural net should relate to the structure of the generative model. References [1] and [2] both present different different takes on this problem. I think the problem / model class under consideration by the authors has three some interesting properties, which I think they could discuss more explicitly 1. A partially generated image can be compared against the target image at any time. This means that the likelihood at any resampling step is actually a pretty good heuristic for the likelihood of the completed image. This is very different the setting in, say, state space models, where you evaluate a model against a previously unseen data point at every resampling step. 2. As far as I can tell, the models under consideration don't have any global parameters. This greatly simplifies the problem. For example, if you want to do sequential inference in a Gaussian mixture model in which you explicitly sample the model parameters, then you would need to design a neural net that predicts the mean and covariance for each cluster given a dataset, which is a highly problem-specific task. 3. Because the authors are considering images, they can extract features in a coarse-to-fine like manner by averaging over pixels. This seems like a great thing to be able to do. 4. Another interesting thing about this setting (which the authors do discuss explicitly) is that the evaluation cost of the neural nets is not negligible. One therefore has to consider not only the amortized cost of training, but also the cost of prediction. In terms of impact, I think the specificity of the problem class considered by the authors is both a good thing and a bad thing. On the one hand you might argue that there is little here that generalizes beyond procedural graphics tasks. On the other I think the paper serves as a nice case study. On balance I am inclined to say this paper just about meets the threshold for acceptance. Post Author Feedback edit: While I do agree with the authors that data driven proposals can, in principle, be of great use in vision as inverse graphics approaches, I am not immediately convinced that the particular neural net architecture proposed by the authors is directly applicable to this task. Minor > Our insight is that while such an oracle can be difficult or impossible to write by hand, it is possible to learn a program chain_neural that comes close. - Let's get rid of "Our insight" and cite [1] and [2] here. > When used for procedural modeling, however, it leads to models whose outputs lack diversity, making them unsuitable for generating visually-varied content. - In this context it would be helpful to state SMC is an importance sampling method. In importance sampling methods the target density must be absolutely continuous with respect to the proposal density (see any introductory reference, the first that pops to mind is [3]), which in the authors language equates to the requirement that the proposal density is "less compact" than the target density. - Why do the authors use a Gaussian to define the loss criterion in equation (3)? A priori, something like an exponential would seem more natural. - In section 4.2, I would like the authors to define the notation ||∇ I_2(p)||=1, since a gradient is technically not defined for a binary image. - The authors should call out the SI when positing equation 1. - In the derivation of equation 1 in SI: xs ∼ P_CM(x) -> xs ∼ P_CM(x | c) - The authors should probably cite the Wake Sleep algorithm [4] as related work. References [1] Gu, S., Ghahramani, Z., and Turner, R. E. (2105) "Neural Adaptive Sequential Monte Carlo". Neural Information Processing Systems [2] Paige, B., and Wood, F. (2016) Inference Networks for Sequential Monte Carlo in Graphical Models. International Conference for Machine Learning Research. [3] Geyer, C. J. (2011). Importance Sampling, Simulated Tempering, and Umbrella Sampling. In Handbook of Markov Chain Monte Carlo Methods. [4] Hinton, G. E., Dayan, P., Frey, B. J., and Neal, R. M. (1995). The "wake-sleep" algorithm for unsupervised neural networks. Science, 268(5214):1158–1161.

Confidence in this Review

3-Expert (read the paper in detail, know the area, quite certain of my opinion)


Reviewer 5

Summary

The paper introduces a neurally-guided procedural model. Unlike probabilistic inference algorithms that generally require a large amount of samples, the proposed model highlights its ability in learning how to satisfy constraints. The model achieves a speedup of more than 10 times.

Qualitative Assessment

The paper introduces a neurally-guided procedural model. Unlike probabilistic inference algorithms that generally require a large amount of samples, the proposed model highlights its ability in learning how to satisfy constraints. The model achieves a speedup of more than 10 times. It describes an interesting problem. Presentation is clear, and results demonstrate well the advantages of the model. I suggest the authors to change "neurally-guided" as it is ambiguous, which can emphasize either biological or artificial neural network. Importance sampling seems a core component in reducing computational time. It would be helpful if the authors can elaborate the discussions on computational time, for example, how much of the speed improvement is from importance sampling? how does it depend on the complexity of the network? It would be interesting to discuss how the proposed method could be applied to more natural data in real-world applications. Despite multiple references and applications such as procedural cities, decorative patterns, and circuits design, I could not understand how the proposed work is better than the current practice in those applications. For example, I don’t think the proposed work solves an actual problem in circuit design or useful to make a mask or layout.

Confidence in this Review

2-Confident (read it all; understood it all reasonably well)


Reviewer 6

Summary

Authors use discriminative models (neural networks) to significantly improve proposals ("as an importance sampling") in sequential Monte Carlo for inference in procedural graphics programs. They show on several examples that their approach brings significant improvements for sch inference.

Qualitative Assessment

I have really enjoyed the paper and believe it is an amazing one. I personally believe it is important to marry generative modelling and discriminative modelling, and this paper successfully presents one of ways to do so, provides an illustrative examples, and shows that the inference for generative models is significantly improved by employing discriminative models. It might be interesting, for authors, to check another already existing, similar work, on using neural network proposals for sequential Monte Carlo inference in procedural models (probabilistic programming), for a slightly related problem of object recognition and tracking (should be available on arXiv).

Confidence in this Review

3-Expert (read the paper in detail, know the area, quite certain of my opinion)